# Effects of Meteorological Factors and Air Pollutants on COVID-19 Transmission under the Action of Control Measures

**DOI:** 10.3390/ijerph19159323

**Published:** 2022-07-29

**Authors:** Fei Han, Xinqi Zheng, Peipei Wang, Dongya Liu, Minrui Zheng

**Affiliations:** 1School of Information Engineering, China University of Geosciences, Beijing 100083, China; 2004200030@cugb.edu.cn (F.H.); peipeiwang@cugb.edu.cn (P.W.); dyliu@cugb.edu.cn (D.L.); 2School of Public Administration and Policy, Renmin University of China, Beijing 100872, China; minruizheng@ruc.edu.cn

**Keywords:** COVID-19 incidence rates, spatial autocorrelation, GWR, GeoDetector, interaction

## Abstract

At present, COVID-19 is still spreading, and its transmission patterns and the main factors that affect transmission behavior still need to be thoroughly explored. To this end, this study collected the cumulative confirmed cases of COVID-19 in China by 8 April 2020. Firstly, the spatial characteristics of the COVID-19 transmission were investigated by the spatial autocorrelation method. Then, the factors affecting the COVID-19 incidence rates were analyzed by the generalized linear mixed effect model (GLMMs) and geographically weighted regression model (GWR). Finally, the geological detector (GeoDetector) was introduced to explore the influence of interactive effects between factors on the COVID-19 incidence rates. The results showed that: (1) COVID-19 had obvious spatial aggregation. (2) The control measures had the largest impact on the COVID-19 incidence rates, which can explain the difference of 34.2% in the COVID-19 incidence rates, while meteorological factors and pollutant factors can only explain the difference of 1% in the COVID-19 incidence rates. It explains that some of the literature overestimates the impact of meteorological factors on the spread of the epidemic. (3) The influence of meteorological factors was stronger than that of air pollution factors, and the interactive effects between factors were stronger than their individual effects. The interaction between relative humidity and NO_2_ was stronger. The results of this study will provide a reference for further prevention and control of COVID-19.

## 1. Introduction

On 31 December 2019, the Chinese government first reported the outbreak of COVID-19 in Wuhan, the capital of Hubei Province, and the epidemic spread swiftly from Wuhan to all provinces in China. At the beginning of 2020, COVID-19 quickly swept across the world, forming a new global “international public health emergency”, which had a tremendous impact on the world economy and public safety [1].

Atmospheric conditions are considered an important factor affecting the spread and infection of COVID-19 [2,3,4]. From 4 to 6 August 2020, the World Meteorological Organization held an online international conference to discuss the relationship between environmental meteorological factors and COVID-19, emphasizing that the epidemic situation may be affected by environmental meteorological factors [5]. Since the epidemic of the COVID-19, the relationship between its spread and meteorological factors has attracted much attention [6,7,8]. Research on the relationship between climatic factors and COVID-19 epidemic situation in different provinces of China shows that the temperature in Hubei, Hunan, and Anhui provinces is positively correlated with COVID-19 spread. There is a negative correlation between Zhejiang Province and Shandong Province; Guangdong Province, Henan Province, Jiangxi Province, Jiangsu Province, and Heilongjiang Province have a mixed relationship of positive and negative correlation [9]. The transmission speed of COVID-19 in Beijing, Shanghai, Guangzhou, and Chengdu is negatively correlated with air temperature, relative humidity, and absolute humidity [10]. Under the condition of controlling population migration, meteorological factors play an independent role in the transmission process of COVID-19 [11]. Xie et al., analyzed 122 cities in China by applying the generalized additive model and found that when the average temperature is lower than 3 °C, there is a positive linear correlation with the incidence of COVID-19. The daily number of confirmed cases can increase by 4.861% when the temperature rises by 1 °C, but the relationship tends to be flat when the temperature rises above 3 °C, [12]. A study based on 224 cities in China shows that temperature does not affect the spread of COVID-19 [13]. The research on the relationship between climatic factors and COVID-19 in the United States shows that the average temperature, minimum temperature, and air quality are significantly related to COVID-19 transmission. However, there is no evidence that climate warming can inhibit COVID-19 transmission [2]. The research in India shows that meteorological factors are the important factors that determine the incidence of COVID-19, among which the highest, lowest and average temperatures are related to COVID-19. There is a significant positive correlation between the average temperature in Jakarta, India, and the incidence of COVID-19 [14]. However, according to the research on the relationship between the temperature and the COVID-19 epidemic in Spain, there is no evidence of a relationship between them [15]. The literature reaffirms the inhibition of temperature on the spread of COVID-19 according to samples from Brazil [16]. Jahangiri et al., have found contradictory results [17]. Therefore, the influence of temperature on the COVID-19 epidemic situation needs to be further studied. Humidity is also a crucial contributor to the transmission of COVID-19 [6,18]. The influence of wind on COVID-19 transmission is controversial in the empirical literature, ranging from positive [19] to negative [20], to insignificant [2].

Air pollutants are another crucial atmospheric factor affecting COVID-19 transmission [21,22]. Air pollution aggravates the impact of COVID-19 on people, which may not only increase the number of infected people but also aggravate COVID-19 symptoms and increase the risk of death [23]. Zhu et al., utilized the generalized additive model to study the relationship between particulate matter and the daily number of COVID-19 confirmed cases in 120 cities in China and found that short-term exposure to PM_2.5,_ PM_10_, CO, NO_2__,_ and O_3_ is positively correlated with the number of COVID-19 confirmed cases, while exposure to SO_2_ could decrease the risk of COVID-19 infection [24]. Many researchers have demonstrated that COVID-19 infections increase in a person exposed to air pollution for both long and short periods [3,24]. A study in Wuhan, Huanggang, and Xiaogan, Hubei province, where the epidemic situation is difficult, found that the average daily incidence of COVID-19 positively correlates with PM_2.5_, while PM_10_ is negatively correlated with COVID-19 [25].

At present, many studies have analyzed the temporal and spatial evolution of the epidemic situation. In foreign countries, they investigated the distribution of the COVID-19 epidemic at the provincial scale in Iran and the county scale in the United States [26,27]. In China, many scholars have conducted a number of research on the spatiotemporal evolution characteristics of the COVID-19 epidemic at the national [28], provincial [29], and municipal scales [30]. These studies reveal that the COVID-19 epidemic has significant agglomeration characteristics at the provincial, municipal, and street scales.

Through analysis, it is found that there are three problems in the current research on atmospheric environment and COVID-19: 1. The existing studies have only discussed its spatial characteristics based on epidemiology, and the research from the perspective of geography needs to be further improved. 2. The correlation between average temperature and the COVID-19 epidemic situation is positive or negative, but there is no consistent relationship; 3. Most studies only focus on the influence of a single atmospheric environmental factor on COVID-19 but do not explore the interaction between atmospheric environmental factors and COVID-19.

To solve the above problems, firstly, the Geoda software 1.20 (Dr. Luc Anselin and his team [31], Urbana, IL, USA) was used to analyze the spatial autocorrelation of the cumulative confirmed cases of COVID-19, and Moran’s I scatter diagram and LISA cluster diagram were drawn to analyze the spatial characteristics of COVID-19 in China. Then, the generalized linear mixed effect model (GLMMs) was used to evaluate the explanation degree of COVID-19 incidence rates variation caused by City-specific characteristics, control measures, meteorological factors, and pollution factors. In addition, the geographically weighted regression model (GWR) was used to analyze the impact of meteorological factors and air pollutants on the COVID-19 incidence rates. Finally, the geological detector (GeoDetector) was introduced to explore the influence of interactive effects between factors on the COVID-19 incidence rates.

## 2. Materials and Methods

### 2.1. Study Area

In this study, we used the cumulative confirmed cases of COVID-19 in China from 24 January 2020 (Wuhan closed city) to 8 April 8 2020 (the government lifted the travel limitations) [32]. The research period of this paper is from January to April 2020. During this period, Hubei province was the “disaster area” of the epidemic. Every day, there were a large number of confirmed cases in Hubei province, the epidemic situation was out of control, and there were a large number of undocumented cases [33]. The COVID-19 epidemic situation in Hubei exhibited different spatial transmission characteristics and temporal dynamic changes [34], so we excluded Hubei Province. In addition, ArcGIS Pro 2.8.3 (Esri Company, Redlands, CA, USA). was used to draw cumulative confirmed cases of COVID 19 in China, as shown in Figure 1.

### 2.2. Data Sources

There are four main types of data in this research: (1) From 24 January to 8 April 2020, the daily COVID-19 confirmed cases were released on the official website of each city’s health commission. To rule out the effect of population density, we selected daily COVID-19 incidence rates. (2) Air pollutants data including particles with diameters ≤ 2.5 μm (PM_2.5_), particles with diameters ≤ 10 μm (PM_10_), and nitrogen dioxide (NO_2_) were obtained from the China National Environmental Monitoring Centre (http://www.cnemc.cn/ (accessed on 10 April 2020)). (3) Meteorological data, including daily average temperature, relative humidity, and wind speed, were provided by the National Meteorological Information Center (http://www.cma.gov.cn/ (accessed on 10 April 2020)). (4) Control measures are represented by the stringency index. The stringency index was provided by Our World in Data (https://ourworldindata.org/covid-stringency-index (accessed on 7 May 2022)). The nine metrics used to calculate the stringency index are: closures of public transport; school closures; cancellation of public events; workplace closures; restrictions on public gatherings; stay-at-home requirements; public information campaigns; restrictions on internal movements; and international travel controls. The index for any given date is calculated as an average score for nine metrics, each with a value between 0 and 100.

### 2.3. Research Methods

Firstly, the generalized linear mixed effect model (GLMMs) is used to analyze to what extent the variability of COVID-19 is attributed to the city-specific characteristics, meteorological factors, air pollution factors, and the control measures at the city level, in order to evaluate the impact of meteorological factors and air pollution factors on COVID-19 after the implementation of control measures. Secondly, the Spearman correlation analysis of meteorological factors, air pollution factors, and COVID-19 incidence rates is carried out, and the significant related influencing factors are screened out at the 0.05 significance level. Thirdly, the geographically weighted regression model (GWR) is used to analyze the significant correlation factors in Spearman and the COVID-19 incidence rates. Finally, the geographical detector (Geodetector) is used to explore the explanation of the interaction between factors on the COVID-19 incidence rates.

#### 2.3.1. Statistical Analysis

We evaluated the extent to which the variability of COVID-19 can be attributed to city-specific characteristics, meteorological factors, air pollution factors and stringency index at the city level. We used the generalized linear mixed effect model (GLMM) to test the explanation degree of five schemes composed of different variables to the variability of COVID-19 incidence rates. We denote *y_ij_* as the daily incidence on day *j* in city *i*, following a Poisson distribution with mean *λ_ij_*. The M5 scheme form is as follows: (1)$$\ln\left(\lambda_{ij}\right)=\beta_{0}+\sum_{p}\beta_{p}x_{p_{i}}+\sum_{q}\beta_{q}x_{q_{ij}}+\beta_{d}x_{d_{ij}}+\beta_{t}\;\mathrm{time}\;e_{i}+\alpha_{i}\;$$
where *β*_0_ is the grand intercept, $x_{p_{i}}$ is the *p*-th city-specific characteristic variable of city *i* with regression coefficient *β_p_*, $x_{q_{ij}}$ is the *q*-th time-varying meteorological and air pollution variables of city *i* on day *j* with regression coefficient *β_q_*, variable time *e_i_* is a time trend variable which is the number of days since the date of the first case with illness in city *i* with regression coefficient *β_t_* in the model, $x_{d_{ij}}$ is the variable with regression coefficient *β_d_* which captures the incremental effect of city-specific stringency index implemented on day *k*. $x_{d_{ij}}$ is defined as below: (2)$$\;x_{d_{ij}}=\left\{\begin{array}{c} 0\;\mathrm{where}\;j\leq \mathrm{date}\;\mathrm{of}\;\mathrm{stringency}\;\mathrm{index}\;\mathrm{implementation}\\ j-k\;\mathrm{where}\;j>\;\mathrm{date}\;\mathrm{of}\;\mathrm{stringency}\;\mathrm{index}\;\mathrm{implementation}\; \end{array}\right.$$

In the GLMM, the city-specific random effect is modeled as *α_i_* which follows a normal distribution with mean 0 and variance *σ_α_*^2^. The use of the random effect is to capture the city-specific heterogeneity that cannot be accounted for by our data.

We compared five schemes: the M1 scheme only has time trend, the M2 scheme has city-specific characteristics and time trend, the M3 scheme has city-specific characteristics, meteorological factors, air pollution factors, and time trend, the M4 scheme has city-specific characteristics, stringency index variables and time trend, the M5 scheme has city-specific characteristics, meteorological factors, air pollution factors, stringency index and time trend (Figure 2). R^2^ proposed by Nakagawa and colleagues [35] is used to determine which scheme can best explain the variance of COVID-19 incidence rates. We use R^2^*_fixed_* to describe the proportion of variance explained by the fixed effect, and R^2^*_random_* to describe the proportion of variance explained by the random effect of urban heterogeneity. The relative risk (RR) and the corresponding 95% confidence interval (CI) are used to quantify the influence of these variables on COVID-19.

#### 2.3.2. Geographically Weighted Regression (GWR)

A geographically weighted regression model (GWR) is an expansion of the ordinary linear regression model, which integrates the spatial characteristics of the data into the regression model so that the model has both linear regression characteristics and spatial correlation [36]. Compared with the traditional least squares (OLS) regression model, the geographically weighted regression model can estimate the influence degree of the explanatory variables in different spatial locations, reflecting the spatial stationarity of the parameters in different regions.

Some studies have confirmed that the distribution of COVID-19 shows spatial heterogeneity [37,38,39], so this study used the geographically weighted regression model (GWR) to estimate the degree of influence of the explanatory variables on COVID-19 incidence in different spatial locations, the COVID-19 incidence rates $y_{i}$
(3)$$\;y_{i}=\beta_{0}\left(u_{i},v_{i}\right)+\sum_{k=1}^{n}\beta_{k}\left(u_{i},v_{i}\right)x_{ik}+\varepsilon_{i}\;i=1,\;2,\;n\;$$
where, $y_{i}$ is the COVID-19 incidence rates of the city *i* at location $\left(\mu_{i}\right.\left.,v_{i}\right)$; $\;\beta_{k}\left(u_{i},v_{i}\right)$ is the *k*-th local regression coefficient value at the *i*-th sampling point, which is a function of the geographic location, depending on the spatial weight matrix; The local regression coefficient indicates the degree of influence of explanatory variables on COVID-19 in different spatial positions. $\;\beta_{0}$ means intercept at a special position $\left(\mu_{i}\right.\left.,v_{i}\right)$, $x_{ik}$ denotes PM_2.5_, PM_10_, NO_2_, relative humidity, temperature, and wind speed, respectively. The location $\left(\mu_{i}\right.\left.,v_{i}\right)$ represents the central coordinates of city i, and $\varepsilon_{i}$ represents the error term of sample i. In this paper, the geographic weighting tool in ArcGIS Pro 2.8.3 is used to build the model, and the best parameter settings are obtained after many experiments and comparisons. Finally, the fixed distance method of the Gaussian kernel function is selected to create a smoother kernel surface. The specific formula is as follows
(4)$$W_{ij}=\exp\left[-{\left(\frac{d_{ij}}{b}\right)}^{2}\right]\;$$
where: $W_{ij}$ is the weight between the i region and the *j* region; $d_{ij}$ is the distance between the *i* region and the *j* region; b is the bandwidth coefficient, which means that the farther the distance is, the slower the weighted attenuation is. An appropriate bandwidth coefficient is very important for analyzing the GWR model because the estimation deviation of the regression coefficient will increase if the bandwidth variable is too small. In order to make the estimation effect of the local regression coefficient better, this research adopts the Akaike information criterion (AIC) to select the optimal bandwidth parameters.

#### 2.3.3. Geographic Detector Model (Geodetector)

The geographical detector can detect the spatial divergence of the distribution of geographical things or phenomena, revealing the driving factors behind this divergence [40]. This method was first used to explore the mechanism of influence of disease risk. In this study, the Geodetector was used to study the driving mechanism behind the COVID-19 incidence rates and the influence of interactive effects between factors on the COVID-19 incidence rates [40]. Factor detection was used to detect the spatial differentiation of COVID-19 incidence rates and the explanatory power of different influencing factors on COVID-19 incidence rates [41]. Interaction detection was used to detect whether the joint effect of two or more influence factors on the COVID-19 incidence rates is significantly greater or less than the independent effect of a single influence factor and whether the influence of these factors on the COVID-19 incidence rates is independent [40], and the interaction relationship is determined by the location of q (x ∩ y) in the seven intervals (see Table 1). *q* is an indicator to measure the explanatory power of independent variables to COVID-19 incidence rates. The formula for *q*-value is as follows:(5)$$\begin{array}{c} \;q=1-\frac{\sum_{h=1}^{L}N_{h}\sigma_{h}^{2}}{N\sigma^{2}}=1-\frac{SSW}{SST}\\ \;SSW=\sum_{h=1}^{L}N_{h}\sigma_{h}^{2},SST=N\sigma^{2} \end{array}$$
where, *L* is a classification of COVID-19 incidence rates or explanatory variables; *N* is the number of sample units in the study area, and $N_{h}$ is the numbers of units in layer *h*; $\;\sigma^{2}$ and $\sigma_{h}^{2}$ and are the variance of the *Y* value of study area and layer *h*, respectively. *SSW* and *SST* are the sum of intra-layer variance and the total variance of the study area, respectively. *q* ∈ [0, 1], the larger the value of q, the stronger the explanation of independent variables to dependent variables. This model is freely available. Free utility sharing software from (http://geodetector.cn/ (accessed on 7 May 2022)).

## 3. Results

### 3.1. Global Spatial Autocorrelation Analysis

To investigate the spatial autocorrelation of COVID-19 propagation, Moran’s I index was used to estimate the interaction among cities in China. According to Moran’s I scatter plot, the correlation between the cumulative confirmed cases and the spatial lag vector can be retrieved. Calculated by Geoda software [42], Figure 3 and Figure 4 show Moran’s I scatter diagram and Z score plot of the cumulative confirmed cases. As seen from Figure 3, on 8 April 2020, the global Moran’s I = 0.129, the cumulative confirmed cases demonstrated positive spatial autocorrelation and “agglomeration” characteristics.

As can be seen from Figure 4, under the significance level of 0.05, Moran’s I of the cumulative confirmed cases on 8 April was 0.129, and z = 6.7610, z > 1.96, *p* = 0.0001, *p* < 0.05, which had statistical significance, indicating that the positive spatial autocorrelation of epidemic spread in the whole country.

### 3.2. Local Spatial Autocorrelation Analysis

The Geoda software was used to conduct spatial local autocorrelation analysis on the cumulative confirmed cases, and the LISA agglomeration map was drawn (Figure 5). As seen from Figure 5, the level of cumulative confirmed cases varied largely among cities in China, which was in a state of polarization. The spatial distribution of cumulative confirmed cases in different provinces showed agglomeration: (a). The cumulative confirmed cases in central provinces and neighboring provinces were large, and the difference of the non-cumulative confirmed cases was small (HH); (b). The cumulative number of confirmed cases in central and neighboring provinces was small, and the difference of cumulative confirmed cases was small (LL); (c). The cumulative confirmed cases in central provinces were large, while that in surrounding provinces was small, and the cumulative confirmed cases varied greatly (HL); (d). The cumulative confirmed cases in central provinces were relatively small, while the cumulative confirmed cases in surrounding provinces were relatively large, with a large difference in the cumulative confirmed cases (LH). The cumulative confirmed cases were HL in Sichuan province, HH in Hubei, Hunan, Jiangxi, Anhui, Shandong, Chongqing, and Henan provinces, and LL in Xinjiang and Qinghai provinces. Local Moran’s I was not significant in other provinces.

### 3.3. Descriptive Analysis

It can be seen from Table 2 that the average population was 6.05 million; the population density ranged from 66 people/km^2^ to 6523 people/km^2^. GDP per capita was 22 thousand to 190 thousand Chinese yuan. Beijing had the highest proportion of people with higher education (42.3%), while Chongqing had the highest proportion of people over 64 (12.9%). In addition, Table 2 also illustrates the statistical indicators of air pollution and meteorological variables. During the observation period, the daily average concentrations of PM_2.5_, PM_10_, and NO_2_ in this study were 46.43 μg/m^3^, 62.97 μg/m^3^, and 19.28 μg/m^3^. The average values of daily average temperature, relative humidity, and wind speed were 2.82 °C, 67.25%, and 2.11 m/s, respectively. Stringency Index ranged from 5.56 to 79.17.

### 3.4. The Generalized Linear Mixed Effect Model (GLMM) Analysis Results

Figure 6 shows that temperature, relative humidity, wind speed, PM_2.5_, PM_10_ and NO_2_ were significantly correlated with the COVID-19 incidence rates (temperature: RR = 0.984, 95% CI: 0.969–0.999; relative humidity: RR = 0.993, 95% CI: 0.988–0.997; wind speed: RR = 0.958, 95% CI: 0.939–0.976; PM_2.5_: RR = 0.983, 95% CI: 0.968–0.998; PM_10_: RR = 0.985, 95% CI: 0.969–0.999; NO_2_: RR = 0.967, 95% CI: 0.939–0.996). No significant correlation was found between city characteristics and the COVID-19 incidence rates.

Figure 7 shows the comparison results of five schemes. Compared with M1, M2 with added city-specific characteristics can only explain 4.2% of the COVID-19 incidence rates variation. Based on the M2 scheme, adding meteorological factors and air pollutants as time-varying fixed effects to M3 can explain the 11.8% variation of the COVID-19 incidence rates. However, by adding control measures to the M4 scheme, the 45% variation of the COVID-19 incidence rates could be explained. The M5 scheme added meteorological factors and air pollutants based on the M4, and the explanation degree of the COVID-19 incidence rates variation was only 1% higher than the M4 scheme. By comparing and analyzing the results of the five schemes, it can be found that the control measures significantly impacted the COVID-19 incidence rates, which can explain the 34.2% (R^2^*_fixed_* of M5 minus R^2^*_fixed_* of M3) variation of the COVID-19 incidence rates. On top of this effect, including the meteorological factors and environmental pollutants in the M5 only resulted in 1% increase in the explained variance even though temperature, relative humidity, wind speed, PM_2.5_, PM_10_, and NO_2_ showed a statistically significant association with the incidence rate of COVID-19 (Figure 6).

### 3.5. The Effect of Meteorological Factors and Air Pollutants on COVID-19 Incidence Rates

#### 3.5.1. Spearman Correlation Analyses

From the above results, it can be seen that the stringency index greatly influences the incidence of COVID-19, but the meteorological factors have little influence. As the influence of temperature on COVID-19 incidence is controversial at present, we will analyze the influence of meteorological factors and air pollution factors on COVID-19 incidence in detail. Spearman correlation coefficient is used to select the influencing factors of the COVID-19 epidemic situation in southern and northern China. In this paper, the correlation analysis is carried out under the condition that the significance level is 0.05. There is no correlation between the correlation coefficients of 0–0.09, with 0.1–0.3 as weak correlation, 0.3–0.5 as medium correlation, and 0.5–1 as strong correlation [43]. 

As shown in Figure 8a, temperature and relative humidity in northern cities were negatively correlated with the COVID-19 incidence rates. There was no obvious correlation between wind speed and COVID-19 incidence rates. Air pollution indicators were all negatively correlated to the COVID-19 incidence rates at a 5% significance level. In southern cities, temperature presented a positive correlation to COVID-19 incidence rates. On the other hand, relative humidity and wind speed were significantly negatively correlated to the COVID-19 incidence rates. Furthermore, all air pollution indicators were negatively correlated with the COVID-19 incidence rates (Figure 8b). 

#### 3.5.2. Spatial Differences of Various Factors Based on GWR Model Analysis

Spearman correlation analysis was carried out on the six influencing factors and COVID-19 incidence rates. Through the 0.05 significant difference test, the influencing factors in the northern region are temperature, relative humidity, PM_2.5_, PM_10_, and NO_2,_ while those in the southern region are temperature, relative humidity, wind speed, PM_2.5_, PM_10_, and NO_2_. Using ArcGIS Pro 2.8.3 to do GWR analysis on these factors and COVID-19 incidence rates, all of them have a good fitting effect (The overall R^2^ in the northern region reached 0.75, and the overall R^2^ in the southern region reached 0.61)), and the spatial heterogeneity is significant.

Figure 9 and Figure 10 demonstrate that the influence of air pollutants and meteorological factors on the COVID-19 incidence rates was spatially heterogeneous. Among them, the influence of temperature, relative humidity, and PM_10_ on the epidemic situation in northern China is mainly zonal differentiation, with the smallest influence on northeast China and the largest influence on northwest China (Figure 9a,b,d). The influence of PM_2.5_ and NO_2_ on the epidemic situation in northern China decreases gradually from northeast to northwest, with the northeast region being the most affected and the northwest region being the least affected (Figure 9c,e). The influence of temperature, PM_2.5_, and PM_10_ on the epidemic situation in southern China gradually decreases from southeast to southwest. That is, the southeast is the most affected and the southwest is the least affected (Figure 10a,d,e). The influence of wind speed and relative humidity on the epidemic situation in southern China increases from east to west (Figure 10b,c). The influence of NO_2_ on the epidemic pattern in southern China has obvious zonal differentiation characteristics, and the degree of influence decreases gradually from north to south, with the northern region being the most affected and the southern region being less affected (Figure 10f).

As shown in Figure 8, temperature, relative humidity, PM_2.5_, and NO_2_ revealed a positive impact on COVID-19 incidence rates in northern cities (Figure 9a–c,e). Contrarily, PM_10_ adversely affected the COVID-19 incidence rates across northern cities (Figure 9d). In southern China, temperature, PM_10_, and NO_2_ illustrated a positive effect on COVID-19 incidence rates (Figure 10a,e,f). Nevertheless, relative humidity, wind speed, and PM_2.5_ presented a passive influence on COVID-19 incidence rates (Figure 10b–d).

### 3.6. Interaction of Affecting Factors

Single-factor detection and multi-factor interactive analysis of meteorological and pollution factors were carried out on the COVID-19 incidence rates of cities in China using the geographic detector. In northern cities, the factor detector was used to determine the influence degree of each influencing factor on the COVID-19 incidence rates. It can be seen from Figure 11a that, the *q* values of all influencing factors ranged from 0.233 to 0.595, and the magnitude of the significant difference was as follows: temperature (0.595) > relative humidity (0.544) > NO_2_ (0.343) > PM_10_ (0.243) > PM_2.5_. The influence of temperature and relative humidity on the change in COVID-19 incidence rates ranked the first two, indicating that meteorological factors played a leading role in the change in COVID-19 incidence rates. Secondly, NO_2,_ which covers all aspects of human activities, including industrial activities and transportation resources, so the change in NO_2_ concentration can quantify the intensity of human activities, which can reflect the impact of human activities on COVID-19 incidence rates to a certain extent. PM_10_ and PM_2.5_ also had a great influence on the COVID-19 incidence rates.

From the perspective of factor interaction, as shown in Table 3, there are two types of COVID-19 epidemic in northern cities: one was nonlinear enhancement and two was a bilinear enhancement. In addition, the interaction of all two factors was stronger than that of a single factor. For example, the *q* value of the interaction between relative humidity and NO_2_ on the change in COVID-19 incidence rates was 0.991, which had the strongest influence on the COVID-19 incidence rates. Moreover, the interaction between relative humidity and all other influencing factors had a significantly greater effect on the COVID-19 incidence rates than the single effect of relative humidity on the COVID-19 incidence rates (0.544). In addition, compared with other factors, PM_2.5_ had the smallest contribution to the COVID-19 incidence rates, the interaction type with temperature and PM_10_ was a bilinear enhancement, and the interaction with other influencing factors had a non-linear synergistic effect on the COVID-19 incidence rates. Compared with the interaction of other factors, the *q* values of the average temperature, PM_2.5_, PM_10_, and NO_2_ interacting with relative humidity significantly increased, being 0.969, 0.919, 0.979, and 0.991, respectively.

In southern cities, a factor detector was used to determine the impact of each influencing factor on the change in COVID-19 incidence rates. It can be seen from Figure 11b that the *q* value range of all impact factors was 0.051~0.907, and the magnitude of the significant difference was as follows: relative humidity (0.274) > NO_2_ (0.155) > temperature (0.149) >PM_2.5_ (0.059) > PM_10_ (0.051) = wind speed (0.051). The influence of relative humidity on the change in COVID-19 incidence rates ranked first, indicating that relative humidity played a leading role in the change in COVID-19 incidence rates; the second was NO_2_, which covers all aspects of human activities, indirectly indicating that human activities had a larger impact on the COVID-19 incidence rates; the contribution of PM_10_ and wind speed to the change in COVID-19 incidence rates was the smallest, *q* was less than 10%.

An interaction detector was used to detect the effect of interaction between factors on the COVID-19 incidence rates. It can be seen from Figure 11b that the explanatory power of the interaction between factors on the COVID-19 incidence rates was higher than that of a single factor on the COVID-19 incidence rates. For example, the *q* value of NO_2_ interacting with relative humidity on the change in COVID-19 incidence rates was 0.907, which had the strongest impact on the change in COVID-19 incidence rates. The impact of NO_2_ interacting with all other influencing factors on the COVID-19 incidence rates was significantly stronger than that of NO_2_ alone on the COVID-19 incidence rates (0.155). In addition, compared with other factors, PM_10_ and wind speed made the smallest contribution to the COVID-19 incidence rates, but the interaction type of wind speed and PM_10_ was bilinear enhancement (see Table 4). High wind speed may significantly enhance the effect of PM_10_ on the COVID-19 incidence rates. A similar result was acquired for PM_2.5_ and PM_10_. According to Figure 11b, PM_2.5_ and PM_10_ had relatively low explanatory power (*q* = 0.059 and 0.051, respectively). After the interaction of PM_2.5_, PM_10__,_ and NO_2_, the *q* values of the influence on the COVID-19 incidence rates increased significantly, which were 0.836 and 0.838, respectively.

## 4. Discussion

Through research, this paper found that control measures were the main factors affecting the COVID-19 transmission, and meteorological factors had a very small impact on the spread of COVID-19, which was consistent with the research results of Chong et al., [33]. Published related studies show that air pollutants have a great influence on the COVID-19 incidence rates [44]. In this paper, NO_2_ revealed a positive impact on COVID-19 incidence rates in China, which was consistent with a previous study [24]. It has been reported that PM_10_ was positively correlated with the COVID-19 incidence rates in southern cities, which was consistent with the research results of Zhu et al. [24]. However, PM_10_ was negatively correlated with the COVID-19 incidence rates in northern cities. This result supports the research of Jiang et al. [25]. In addition, in this study, COVID-19 incidence rates were positively correlated with PM_2.5_ in most northern cities with relatively high PM_2.5_ levels, which was consistent with the research results of Jiang et al. [25]. COVID-19 incidence rates were negatively correlated with PM_2.5_ in southern cities, this result was contrary to many studies. Although there is no research to prove this, we suspect that this may be because people have strengthened the protection against high concentrations of PM_2.5_ and tried to avoid exposure to PM_2.5_. In conclusion, before reaching a clear conclusion, it is necessary to further study the impact of PM_2.5_ on COVID-19 transmission.

In this paper, it was found that the correlation between average temperature and COVID-19 was positive or negative, without a consistent relationship, which was consistent with the research results of Alvaro B R et al. [15]. Studies have demonstrated that high temperatures may destroy the cell membrane on the surface of the virus, reducing its stability and infectivity [45]. The impact of temperature on COVID-19 needs to be further studied. It has been found that higher wind speed helps to reduce the number of viruses in the air, thus reducing the possibility of virus transmission, that is, the average wind speed is negatively correlated with COVID-19 [46], which is similar to ours. Relative humidity was negatively correlated with COVID-19 in southern cities, which was consistent with some previous studies [47]. This may be due to the immune system having limited immunity to the COVID-19 virus in low humidity environments. Nevertheless, in northern cities, the COVID-19 incidence rates were positively correlated with relative humidity, consistent with a previous study [48].

Our study has some limitations. First of all, the study time is short, and subsequent studies need a longer period to study the COVID-19 incidence rates and its relationship with influencing factors. Secondly, our study is not globally representative because only Chinese cities are analyzed. Thirdly, the predicted R^2^ value in the geographically weighted regression model is low, which indicates that our research results are statistically insufficient. Further analysis should introduce more explanatory variables and capture other key concurrent factors, such as the number of doctors, medical conditions, sex and age of infected patients, living environment and living standard of patients, and other possible influencing factors. Finally, because of the complexity of distinguishing the influence of each control measure, we use a single variable to express the effect of the control measure. It is necessary to use more indicators of control measures for further modeling research in order to rank the importance of the factors that explain the decline in COVID-19.

## 5. Conclusions

This paper mainly analyzed the spatial characteristics and influencing factors of COVID-19 transmission in China in 2020 at the city scale. The following conclusions were obtained from spatial autocorrelation, geographical detectors, geographically weighted regression, and so on. (1) The epidemic distribution had spatial agglomeration. The Moran’s I scatter diagram and LISA cluster diagram were drawn, and the analysis showed that: the cumulative confirmed cases in each city were clustered in spatial distribution. The cumulative confirmed cases were HL in Sichuan, HH in Hubei, Hunan, Jiangxi, Anhui, Chongqing, and Henan, LL in Xinjiang and Qinghai, and Moran’s I in other provinces were not significant. (2) Temperature, relative humidity, NO_2_, and PM_2.5_ exhibited positive effects on COVID-19 incidence rates in northern cities. Contrarily, PM_10_ was negatively correlated with the COVID-19 incidence rates in northern cities; (3) in southern cities, the COVID-19 incidence rates were negatively correlated with wind speed, relative humidity, PM_2.5_, but positively correlated with PM_10_, NO_2_, and temperature. (4) The influence of meteorological factors was stronger than that of air pollution factors, and the interactive effects between factors were stronger than their individual effects. The interaction between relative humidity and NO_2_ was the strongest, and the maximal *q* value of the interaction was 0.991 (*p* < 0.001); (5) the impacts of meteorological factors and air pollutants on COVID-19 incidence rates showed spatial heterogeneity; (6) the control measures had the largest impact on the COVID-19 incidence rates, which can explain the difference of 34.2% in the COVID-19 incidence rates, while meteorological factors and pollutant factors can only explain the difference of 1% in the COVID-19 incidence rates.

## Figures and Tables

**Figure 1 ijerph-19-09323-f001:**
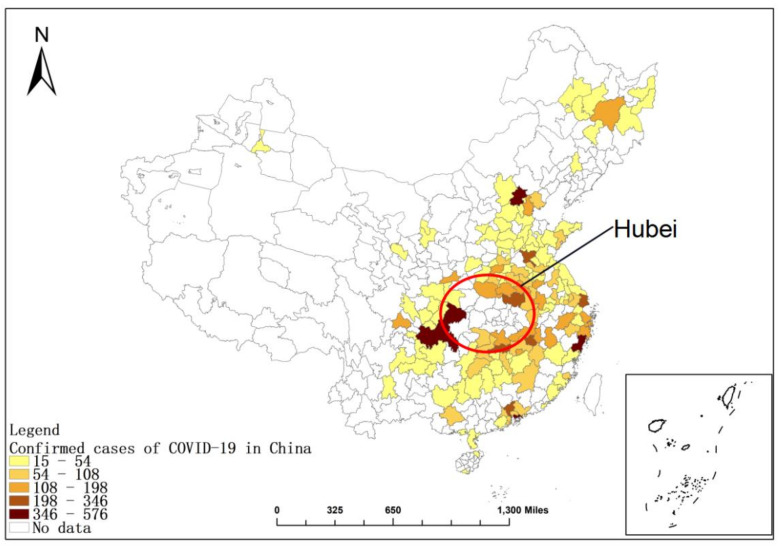
Cumulative confirmed cases of COVID 19 in China. (Note: The base map is from the National Bureau of Surveying and Mapping Geographic Information (http://bzdt.nasg.gov.cn) (accessed on 7 May 2022). We ensure that we have obtained the permission and there is no copyright issue.).

**Figure 2 ijerph-19-09323-f002:**
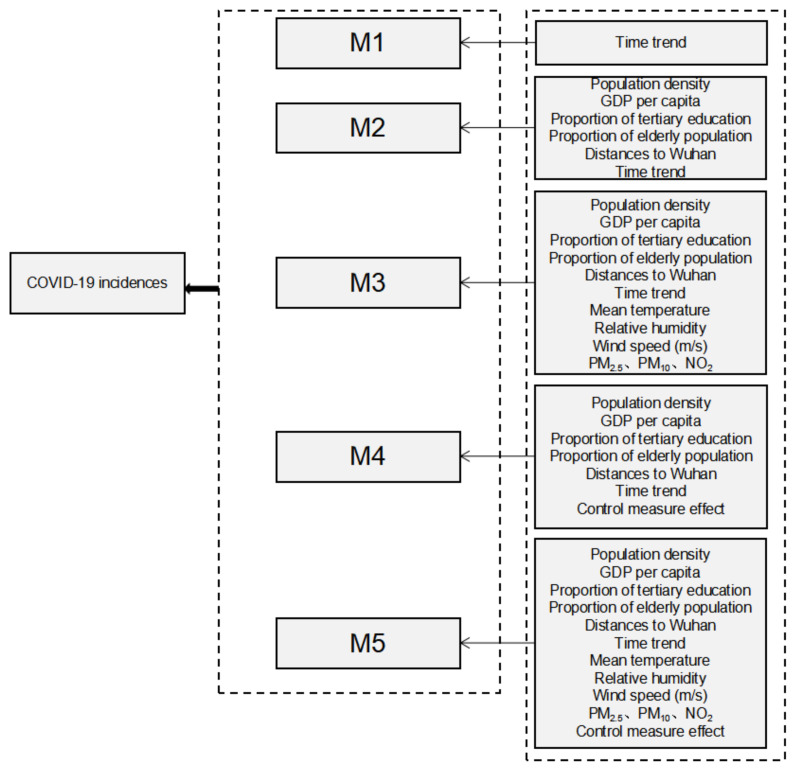
Variables included in the five schemes.

**Figure 3 ijerph-19-09323-f003:**
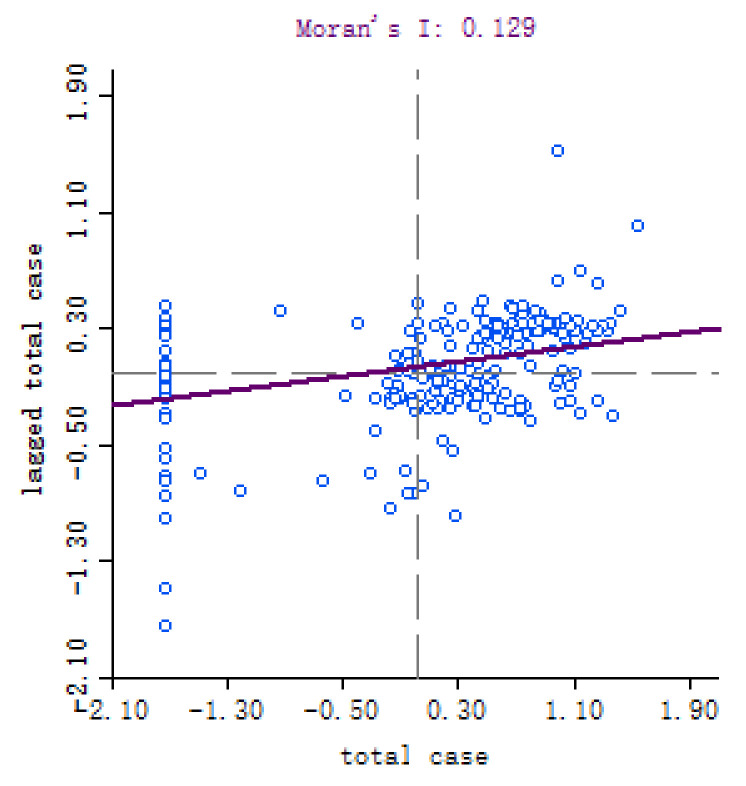
Moran’s I scatter plot of the cumulative number of COVID-19 on 8 April 2020.

**Figure 4 ijerph-19-09323-f004:**
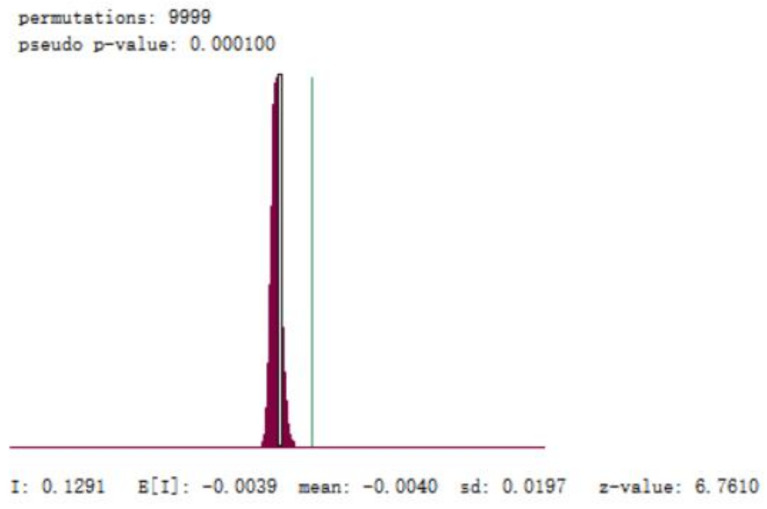
Z-score score chart of the cumulative number of COVID-19 on 8 April 2020.

**Figure 5 ijerph-19-09323-f005:**
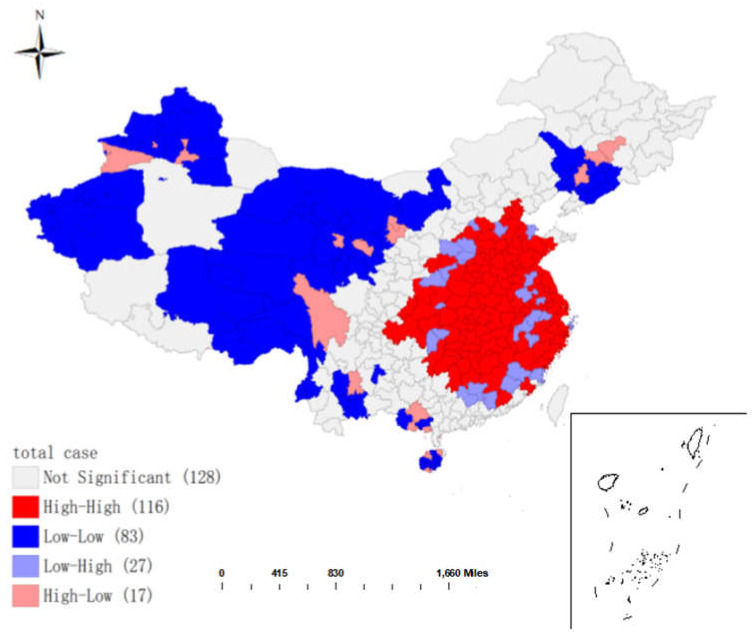
Local Indicator of Spatial Association (LISA) cluster map of the cumulative number of diagnoses on 8 April 2020.

**Figure 6 ijerph-19-09323-f006:**
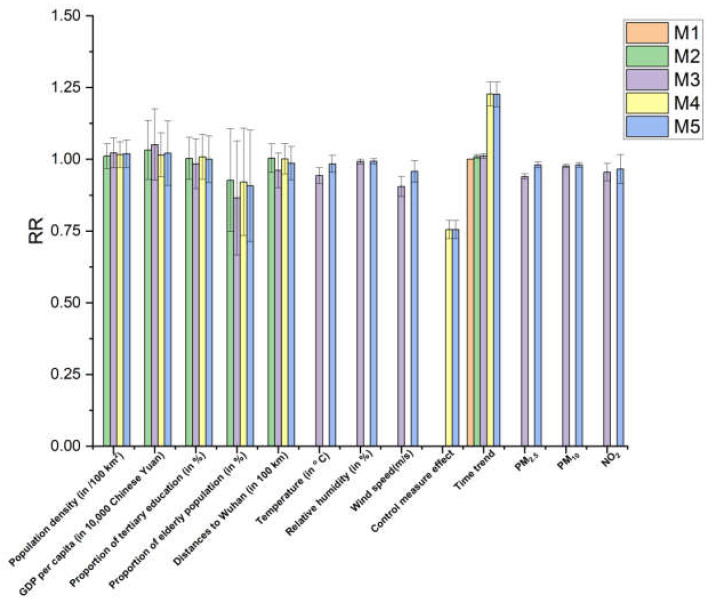
Compare the relative risks (95% confidence intervals) of variables among different schemes (RR: Relative risk in incidence rate of COVID-19 for each unit change in variable).

**Figure 7 ijerph-19-09323-f007:**
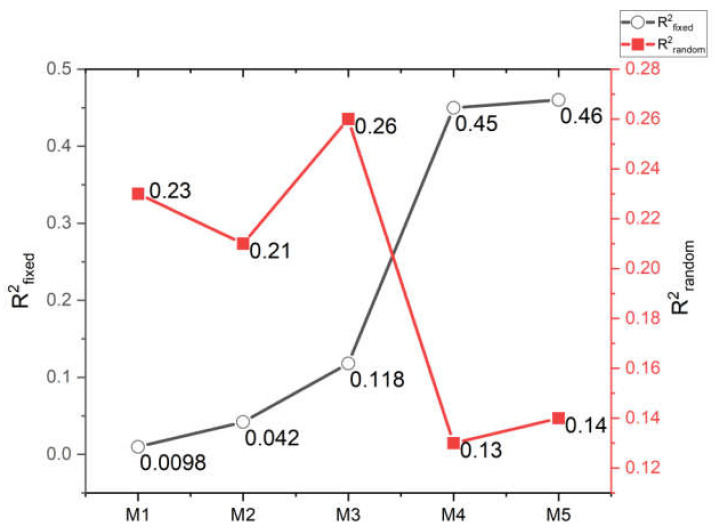
Comparison of changes in R^2^ among different schemes (R^2^_fixed_: Proportion of variance in the COVID-19 incidence rate explained by the fixed effect terms; R^2^_random_: Proportion of variance explained the random effect term of cities’ heterogeneity).

**Figure 8 ijerph-19-09323-f008:**
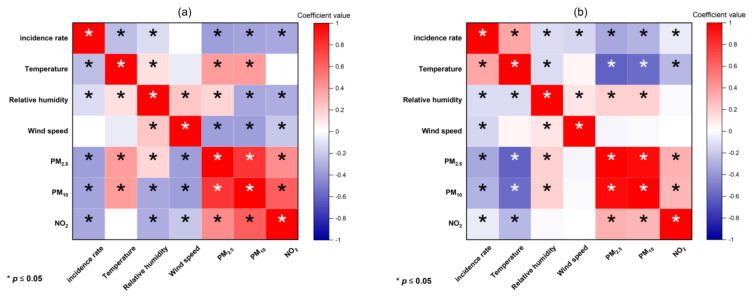
Results of Spearman correlation analysis (* stands for a higher and more significant correlation (*p*-value < 0.05)) in northern cities (**a**) and southern cities (**b**)).

**Figure 9 ijerph-19-09323-f009:**
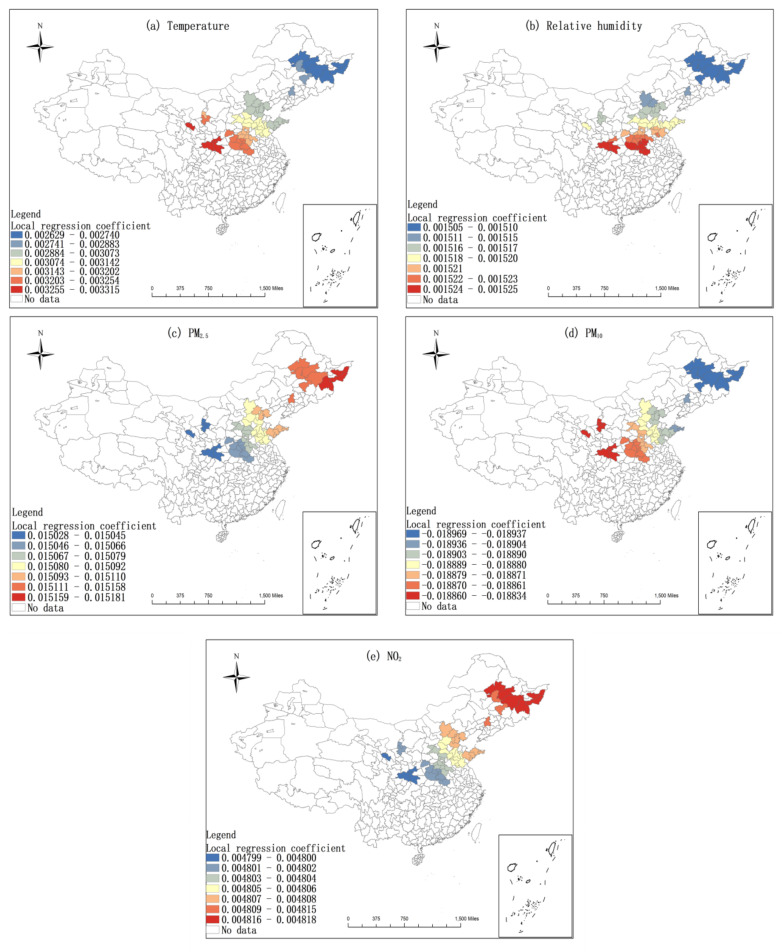
The local regression coefficient diagram of meteorological factors and air pollutants in northern cities.

**Figure 10 ijerph-19-09323-f010:**
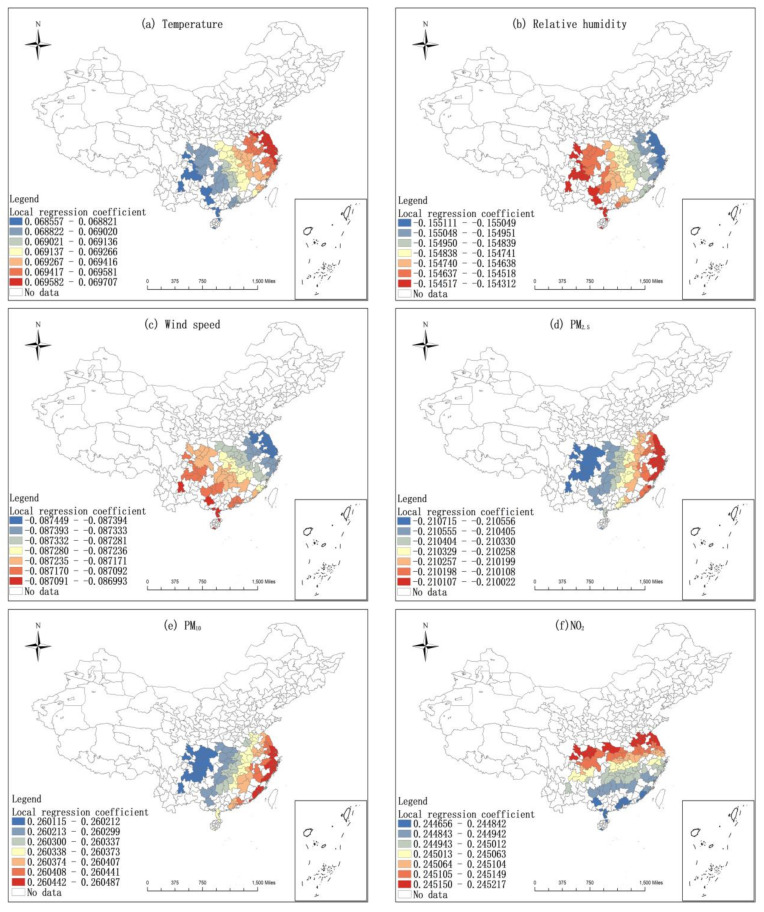
The local regression coefficient diagram of meteorological factors and air pollutants in southern cities.

**Figure 11 ijerph-19-09323-f011:**
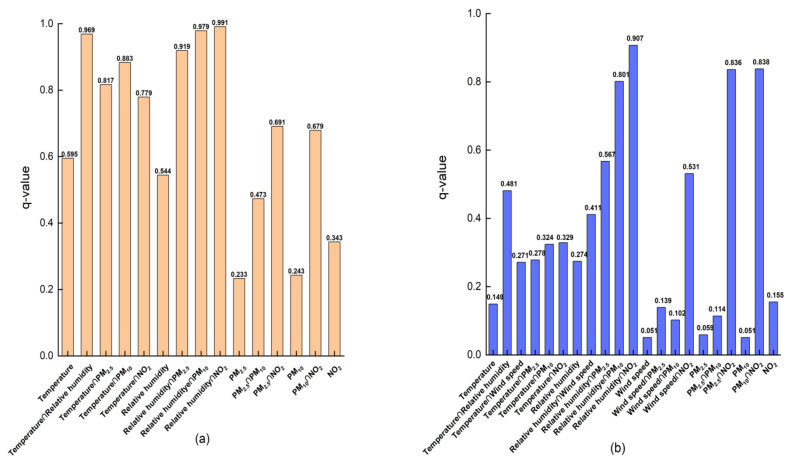
Power of determinants in interaction in northern cities (**a**) and southern cities (**b**).

**Table 1 ijerph-19-09323-t001:** Types of interaction between two covariates.

Interaction Categories	Description
Enhance	*q* (*X*1 ∩ *X*2) > *q* (*X*1) or *q* (*X*2)
Bi-enhance	*q* (*X*1 ∩ *X*2) > *q* (*X*1) and *q* (*X*2)
Enhance, nonlinear	*q* (*X*1 ∩ *X*2) > *q* (*X*1) + *q* (*X*2)
Weaken	*q* (*X*1 ∩ *X*2) < *q* (*X*1) + *q* (*X*2)
Weaken, uni-enhance	*q* (*X*1 ∩ *X*2) < *q* (*X*1) or *q* (*X*2)
Weaken, nonlinear	*q* (*X*1 ∩ *X*2) < *q* (*X*1) and *q* (*X*2)
Independent	*q* (*X*1 ∩ *X*2) = *q* (*X*1) + *q* (*X*2)

**Table 2 ijerph-19-09323-t002:** Descriptive statistics of city-specific characteristics, air pollution, meteorological factors, and stringency index in China.

	Mean	Min	Max
**City-specific characteristics**			
Population (in 10,000)	605	60	3397
Population density (in people/km^2^)	514	66	6523
GDP per capita (in 10,000 Chinese Yuan)	6.3	2.2	19
Proportion of population with higher education (%)	12.0	8.5	42.3
Proportion of elderly population (%)	10.8	7.4	12.9
Distances to Wuhan (in 100 km)	9.3	2.1	32.7
**Meteorological factors**			
Mean temperature (°C)	6.46	−33.8	26.5
Relative humidity (%)	67.26	17	100
Wind speed (m/s)	1.27	0	15.4
**Air pollution**			
PM_2.5_ (μg/m^3^)	39.86	2	554
PM_10_ (μg/m^3^)	53.99	4	632
NO_2_ (μg/m^3^)	16.74	2	86
**Stringency Index**	29.87	5.56	79.17

**Table 3 ijerph-19-09323-t003:** Results of interaction detector in northern cities.

Dominant Interaction Factor	*q* Value	Interact Result
Temperature ∩ Relative humidity	0.969	Bi-enhance
Temperature ∩ PM_2.5_	0.817	Bi-enhance
Temperature ∩ PM_10_	0.883	Enhance, nonlinear
Temperature ∩ NO_2_	0.779	Bi-enhance
Relative humidity ∩ PM_2.5_	0.919	Enhance, nonlinear
Relative humidity ∩ PM_10_	0.979	Enhance, nonlinear
Relative humidity ∩ NO_2_	0.991	Enhance, nonlinear
PM_2.5_ ∩ PM_10_	0.474	Bi-enhance
PM_2.5_ ∩ NO_2_	0.691	Enhance, nonlinear
PM_10_ ∩ NO_2_	0.679	Enhance, nonlinear

**Table 4 ijerph-19-09323-t004:** Results of interaction detector in southern cities.

Dominant Interaction Factor	*q* Value	Interact Result
Temperature ∩ Relative humidity	0.481	Enhance, nonlinear
Temperature ∩ Wind speed	0.271	Enhance, nonlinear
Temperature ∩ PM_2.5_	0.278	Enhance, nonlinear
Temperature ∩ PM_10_	0.324	Enhance, nonlinear
Temperature ∩ NO_2_	0.329	Enhance, nonlinear
Relative humidity ∩ Wind speed	0.411	Enhance, nonlinear
Relative humidity ∩ PM_2.5_	0.567	Enhance, nonlinear
Relative humidity ∩ PM_10_	0.801	Enhance, nonlinear
Relative humidity ∩ NO_2_	0.907	Enhance, nonlinear
Wind speed ∩ PM_2.5_	0.139	Enhance, nonlinear
Wind speed ∩ PM_10_	0.102	Bi-enhance
Wind speed ∩ NO_2_	0.531	Enhance, nonlinear
PM_2.5_ ∩ PM_10_	0.114	Bi-enhance
PM_2.5_ ∩ NO_2_	0.836	Enhance, nonlinear
PM_10_ ∩ NO_2_	0.838	Enhance, nonlinear

## Data Availability

The datasets used and/or analyzed during the current study are not publicly available due to General Data Protection Regulations; however, they are available from the corresponding author upon reasonable request.

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
