# Peer review of "Effects of Meteorological Factors and Air Pollutants on COVID-19 Transmission under the Action of Control Measures"

_ijerph, 2022, doi:10.3390/ijerph19159323_

Round 1

Reviewer 1 Report

The article „Spatial characteristics and influencing factors of COVID 19 epidemic in China” is one of many studies relating to analyses related to the COVID-19 pandemic. The authors used statistical modeling to evaluate the explanation degree of COVID-19 cases variation caused by City-specific characteristics, control measures, meteorological factors and pollution factors. It certainly fits in with current research trends and may be treated as an innovative approach in the analyzed subject matter.

The article is structured correctly, containing all the elements that should be found in a good research paper. I have included comments on the content of each chapter in the comments below.

I ask the authors to respond to the comments below and make any corrections to the article:

1. I ask the authors to organize the statistical elaboration, especially to indicate in a clear unambiguous way for what purpose a step is done, why such and not another method is used, and what the results obtained are used for. In the current version of the article, in the various steps of the statistical elaboration are detached from each other, it is not always explained why they are done.

2. Subsection 2.3.2 - no explanation (which can also be seen in the results) for what purpose GWR was used. For example, there is no statistical results specific to regression, such as R2. The effect of using GWR is basically only generated maps in the article. After all, in this method one can also obtain very important and useful statistical results, which would greatly help in the conducted analysis. The authors used ArcGIS 10.5 software, which is a bit of a limitation as it is no longer the latest. Please use ArcGIS Pro - it gives you much more possibilities, with its use you can basically do all the statistical analysis steps from the article.

3. Fig 7 with a description - what this figure presents, how the authors obtained the sum of 100% by decomposing it into individual schemes. Nowhere in the article is the calculation and interpretation of the R2 indicator methodically explained. Usually in statistical research this indicator shows to what extent the explanatory variable explains the dependent variable. In the article I am not entirely convinced that this is the case. Please provide the calculation and interpretation of the indicator.

4. Subsection 3.5.1 - the given values of correlation coefficients do not authorize the authors to formulate that there is any relationship between the variables. One can even say that there is no correlation between any of the analyzed factors and the number of COVID-19 cases. Please clarify.

5 Fig 9, 10 and their analysis. What it means: The local regression coefficient, what is calculated (missing from the methodology) what values can it take, do the results really explain any impact?

6. Subsection 3.6 - to determine impact and interaction, the authors use R2 once (although I'm not entirely convinced that correctly) once q-value. Why? Why not use the results from GWR and the R2 values obtained there? What values can q-value take and what does it indicate.

7. Fig 12 - unreadable. What do the colors mean, incomplete names, etc.?

8. What do the authors mean by the term: control measures. This is quite an important part of their analysis and nowhere in the article is it described. In the Discussion section you can read that "control measures were the main factors affecting the COVID-19 transmission", in Table 2 - none, in the description - none.

9. I am not a language expert, but in my opinion the article needs linguistic corrections. E.g. I would suggest not using the term: COVID-19 incidences a COVID-19 cases.

10. minor editorial note - with many references to the figures in the article are linked figures on the disk of one of the Authors (fortunately, of course, not to the input...). Please correct this.

11 I would also suggest expanding the scientific background of the study. Currently, you can only see 32 items in the References, most by Chinese Authors. The topic is popular and current all over the world. There are a lot of publications on this topic, please expand the Introduction chapter with more international sources.

Article needs correction.

Reviewer 2 Report

In their manuscript, Han et. al. present an in silico analysis of environmental and public hygiene measures on the spread of COVID-19 in China during the first few months of outbreak to global pandemic. The authors find that control measures are the most associated to spread versus meterological or pollutant factors, similar to other studies. The data from this manuscript is interesting and important; however, the overall presentation and representation needs significant work to increase the novelty and usability of the data.

1)      The manuscript needs an edit for grammar, language, syntax, and tense

2)      It may be better to make the title a bit more specific by indicating that spatial and “influencing factors” deal with, at majority, meteorological factors and air pollutants in this study

3)      Please explain further why Hubei province was excluded from this study. Inclusion of this province is especially important as it is considered to be the epicenter of the pandemic

4)      There is a large amount of interesting data presented in the paper; however, the authors need to focus their manuscript more to explain the analyses and overall outcomes of their reuslts for better understanding. Indeed, it may be better to cut down on the number of figures within the text and only focus on those that have data needed for interpretation and move the rest to supplemental

5)      Figures are extremely hard to read at size and resolution provided

6)      Figure 12 impossible to interpret, especially at greyscale. May be better presented as a table.

7)      Can the authors further comment on how the data was presented temporally? An overall analysis of risk by day might be warranted given the highly variable nature of the data

8)      What control measures are most associated in this study? Could these control measures also be affected by meterological or pollutant factors?

9)      Much further discussion is warranted on the many biases and limitations of the study aside from the ones presented

Round 2

Reviewer 1 Report

I accept the revised version of the article. Thank you for your insightful and complete responses to my comments. I wish you good luck in your future scientific work.

Reviewer 2 Report

The authors have addressed my previous comments to the best of their ability. Thank you.